# RoMA: Robust Model Adaptation
# for Offline Model-based Optimization

**Sihyun Yu**[1]     **Sungsoo Ahn**[2]     **Le Song**[2,3]     **Jinwoo Shin**[1]

[1]Korea Advanced Institute of Science and Technology (KAIST)
[2]Mohamed bin Zayed University of Artificial Intelligence (MBZUAI) [3]BioMap
{sihyun.yu, jinwoos}@kaist.ac.kr
{Peter.Ahn, Le.Song}@mbzuai.ac.ae

## Abstract

We consider the problem of searching an input maximizing a black-box objective function given a static dataset of input-output queries. A popular approach to solving this problem is maintaining a proxy model, *e.g.*, a deep neural network (DNN), that approximates the true objective function. Here, the main challenge is how to avoid adversarially optimized inputs during the search, *i.e.*, the inputs where the DNN highly overestimates the true objective function. To handle the issue, we propose a new framework, coined robust model adaptation (RoMA), based on gradient-based optimization of inputs over the DNN. Specifically, it consists of two steps: (a) a pre-training strategy to robustly train the proxy model and (b) a novel adaptation procedure of the proxy model to have robust estimates for a specific set of candidate solutions. At a high level, our scheme utilizes the *local smoothness prior* to overcome the brittleness of the DNN. Experiments under various tasks show the effectiveness of RoMA compared with previous methods, obtaining state-of-the-art results, *e.g.*, RoMA outperforms all at 4 out of 6 tasks and achieves runner-up results at the remaining tasks.

## 1   Introduction

Designing new objects with the desired property is a fundamental problem in a wide range of real-world domains, including chemistry [13], biology [5, 41], robotics [3, 28], and aircraft design [20]. Unfortunately, many of them often require evaluating an expensive objective function, *e.g.*, synthesizing and measuring the fluorescence of the protein [5]. Model-based optimization (MBO) is a powerful framework to circumvent this issue, based on an approximation of the expensive evaluation of objective functions by a cheap proxy model. Specifically, MBO optimizes a solution with respect to the proxy model instead of the costly objective function. For obtaining a better proxy model, predominant works have typically assumed active (or *online*) learning scenarios, *i.e.*, they allow for updating the model interactively with new queries of choice to approximate the ground-truth objective function better around optima [22, 37, 39, 40, 43, 45, 46].

Recently, researchers have also proposed *offline* versions of MBO algorithms [5, 8, 11, 25, 52] which build the proxy model from a given static dataset, *i.e.*, input-output pairs collected from the objective function prior to building the proxy model. Such frameworks are beneficial for various real-world scenarios where the new interactive access to the objective functions may accompany serious danger or expensive cost for evaluation, *e.g.*, drug discovery or aircraft design. The common challenge to be addressed for offline MBO is to construct a proxy model that yields an accurate approximation of the

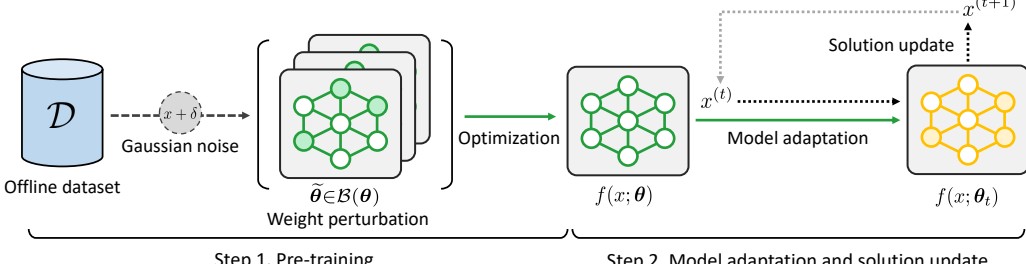

Figure 1: Overall illustration of our proposed robust model adaptation (RoMA) framework.

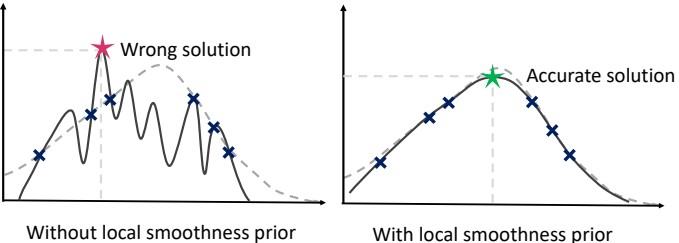

Figure 2: Illustration of the effect of employing local smoothness prior.

true objective function outside the training dataset, as the optimization procedure mostly leads the solution to be out of the dataset and causes severe overestimation due to a domain shift [12, 48].

Intriguingly, Fu and Levine [11], Trabucco et al. [52, 53] found the gradient-based optimization of solutions to be effective for offline MBO. Such a procedure is attractive since it can utilize gradient information of the proxy model, *e.g.*, a deep neural network (DNN), to generate fine-grained solutions. However, due to the highly non-smooth nature of DNNs, such a gradient-based optimization may generate *adversarial examples* [51] whose score deviates significantly between the DNN-based proxy model and the true objective function. Existing works have attempted to regularize this behavior using conservative training [52] or normalized maximum likelihood [11]. In this paper, we investigate a more direct way to regularize such a non-smooth nature of DNN.

**Contribution.** We introduce a new framework, coined robust model adaptation (RoMA), for offline MBO to optimize the solution via gradient-based updates from the DNN proxy model. Our main idea is to utilize a *local smoothness prior* for alleviating the brittleness of DNN at adversarial examples. To be specific, RoMA consists of two steps: (a) a pre-training strategy to robustly train the proxy model and (b) a novel adaptive adjustment procedure of the proxy model to have robust estimates for a specific set of candidate solutions. We provide an overall illustration of our framework in Figure 1.

(a) We first introduce a pre-training procedure of the proxy model to be locally smooth at inputs and eventually becomes robust to the gradient-based input change. The novel component of the pre-training procedure is an additional regularization to force a flat loss landscape with respect to the model parameters, rather than only employing regularization based on input-level perturbation into the training objective. Such consideration of the flatness to the model parameters is effective in improving the robustness of the proxy model to the gradient-based updates.

(b) To strengthen the accurate update of solution candidates from the trained proxy model, we also present a novel adaptive adjustment strategy of the proxy model to enhance the local smoothness at inputs outside the training data. Specifically, we propose to adjust the model at given solution candidates by adding a small perturbation to the model weights. After that, an update of solution candidates occurs from the *adjusted model* rather than the original DNN proxy model. We discover that the proposed adjustment strategy aptly mitigates the fragility of the proxy model at gradient-basead updates to given inputs and improves the performance. We also emphasize that the adaptive strategy is worthwhile in offline MBO as the expressive power of the proxy model is limited under the fixed number of data and parameters; it is difficult to sufficiently regularize all inputs with the domain shift from the dataset without any adjustments.

---

**Algorithm 1** Robust model adaptation (RoMA)

---

1: **for** $k = 0$ to $K$ **do**                     $\triangleright$ ***Step 1.** Pre-training with weight perturbations.*
2:     Find $\widetilde{\boldsymbol{\theta}} \leftarrow \arg\max_{\widetilde{\boldsymbol{\theta}} \in \mathcal{B}(\boldsymbol{\theta})} \mathbb{E}_{(x,y) \sim \mathcal{D}, \delta \sim \mathcal{N}(0,\sigma)} \left[ (f(x + \delta; \widetilde{\boldsymbol{\theta}}) - y)^2 \right]$ via PGD.
3:     Reparameterize $\widetilde{\boldsymbol{\theta}}$ using $\boldsymbol{\theta}$ such that $\widetilde{\theta}_\ell = \theta_\ell + \phi_\ell$ for some fixed $\phi_\ell$ and $\ell = 1, \ldots, L$.
4:     Minimize $\mathbb{E}_{(x,y) \sim \mathcal{D}, \delta \sim \mathcal{N}(0,\sigma)} \left[ (f(x + \delta; \widetilde{\boldsymbol{\theta}}) - y)^2 \right]$ over $\boldsymbol{\theta}$ using gradient-based update.
5: **end for**
6: Set $\boldsymbol{\theta}_{-1} \leftarrow \boldsymbol{\theta}$ and choose $x^{(0)}$ from the training dataset $\mathcal{D}$.
7: **for** $t = 0$ to $T - 1$ **do**            $\triangleright$ ***Step 2.** Optimizing the solution with iterative model adaptation.*
8:     Find $\boldsymbol{\theta}_t \leftarrow \arg\min_{\widetilde{\boldsymbol{\theta}} \in \mathcal{B}(\boldsymbol{\theta})} \left[ \left. ||\nabla_x f(x; \widetilde{\boldsymbol{\theta}})||_2 \right|_{x=x^{(t)}} + \alpha \left( f(x^{(t)}; \widetilde{\boldsymbol{\theta}}) - f(x^{(t)}; \boldsymbol{\theta}_{t-1}) \right)^2 \right]$ via PGD.
9:     Compute $x^{(t+1)}$ from $x^{(t)}$ using gradient-based update.
10: **end for**
11: Output the final solution $x^{(T)}$.

---

We extensively verify the effectiveness of our RoMA under Design-Bench [53], an offline MBO benchmark on diverse domains such as chemistry, biology, and robotics. In particular, our method achieves the best result in 4 out of 6 tasks when compared to strong baselines [5, 8, 11, 25, 52]. For the remaining tasks, RoMA at least achieves the runner-up results. When averaging over all tasks, RoMA achieves the normalized 100th percentile score[1] of $1.705$, while the second-best baseline [11] achieves $1.687$. Intriguingly, our framework dramatically improves the 50th percentile scores, demonstrating how it generates diverse solutions with superior quality.

## 2 RoMA: Robust model adaptation

### 2.1 Overview of RoMA

To describe our framework, we define our problem as follows. Consider a real-valued vector $x \in \mathbb{R}^L$, *e.g.*, protein sequence, and a real-valued objective function $f^*(x) \in \mathbb{R}$, *e.g.*, fluorescence of a protein sequence or the morphology of the robot. We are interested in finding an optimal input $x^* \in \mathbb{R}^L$ that maximizes the objective function, *i.e.*, $x^* = \arg\max_{x \in \mathbb{R}^L} f^*(x)$. Rather than having access to the objective function, we are given a static dataset $\mathcal{D} = \{(x_i, y_i)\}_{i=1}^d$ of input-output pairs $(x_i, y_i)$ queried from the objective function, *i.e.*, $y_i = f^*(x_i)$.

Since the true objective function is unknown, our RoMA framework approximates it using a surrogate model $f(x; \boldsymbol{\theta})$ parameterized as a deep neural network (DNN) with weights $\boldsymbol{\theta}$. Then one can use gradient-based updates to find a fine-grained input maximizing the surrogate model. However, without any regularization, DNNs are prone to overfitting and naïve gradient ascent algorithms can easily find adversarial examples whose score deviates significantly between the surrogate model and the true objective function. Our main contribution lies in resolving this issue by regularizing the smoothness of the surrogate model, *i.e.*, decreasing the sensitivity of the model with respect to input perturbations.

Such a *local smoothness prior* for regularizing DNNs has been successfully used to enhance the generalization of DNNs in various scenarios, *e.g.*, adversarial robustness, generative models, and uncertainty estimation [1, 7, 9, 14, 29, 31, 35, 38, 47, 51, 56, 57]. Especially, we note the striking resemblance between our framework to the adversarial training methods [7, 51, 56]; they seek to prevent the existence of adversarial examples generated from gradient-based optimization of inputs. In this respect, one may expect the smoothness prior to be beneficial for our offline MBO framework.

To be specific, our RoMA is a two-stage framework, described as follows.

**Step 1.** Train the proxy model on the dataset $\mathcal{D}$ to approximate the true objective function with Gaussian smoothing of inputs under worst-case weight perturbations.

**Step 2.** For time-steps $t = 0, \ldots, T - 1$, update the solution $x^{(t)}$ after adapting the output of the proxy model to be locally smooth at the solution $x^{(t)}$.

We provide the detailed description of our framework in Algorithm 1.

---

[1]The percentile score is computed from 128 solutions proposed by the offline MBO algorithms.

In the rest of this section, we explain each component of RoMA in detail. In Section 2.2, we describe how the RoMA pre-trains the surrogate model with Gaussian smoothing (**Step 1.**). In Section 2.3, we explain the procedure of updating the solution while simultaneously adapting the proxy model to yield input-level smoothness at the intermediate solution (**Step 2.**). Finally, in Section 2.4, we describe some other algorithmic components in RoMA, such as an approach to handling discrete inputs or a procedure to select hyperparameters without the true objective function.

## 2.2 Pre-training with weight perturbations

We first describe our pre-training strategy for the DNN-based proxy model $f(x; \boldsymbol{\theta})$ to approximate the true objective function $f^*(x)$ on training dataset $\mathcal{D}$. We focus on regularizing the training with Gaussian smoothing of inputs under worst-case weight perturbations [55]. To this end, we optimize an $L$-layered DNN with weight $\boldsymbol{\theta}$ as the proxy model to minimize the approximation loss as follows:

$$\mathcal{L}(\boldsymbol{\theta}) := \max_{\widetilde{\boldsymbol{\theta}} \in \mathcal{B}(\boldsymbol{\theta})} \left[ \mathbb{E}_{(x,y) \sim \mathcal{D}, \delta \sim \mathcal{N}(0,\sigma)} \left[ (f(x + \delta; \widetilde{\boldsymbol{\theta}}) - y)^2 \right] \right], \tag{1}$$

where $\delta$ is a random variable sampled from the Gaussian distribution $\mathcal{N}(0, \sigma^2)$ with mean $0$ and standard deviation $\sigma$. Furthermore, $\mathcal{B}(\boldsymbol{\theta})$ is the set of perturbed parameters of the DNN that are near the parameter $\boldsymbol{\theta}$, defined as follows:

$$\mathcal{B}(\theta) := \left\{ \widetilde{\boldsymbol{\theta}} : \|\theta_\ell - \widetilde{\theta}_\ell\|_F \leq \varepsilon \cdot \|\theta_\ell\|_F \quad \forall \ell \in \{1, \cdots, L\} \right\}.$$

Here, $\varepsilon > 0$ indicates the maximum magnitude of the perturbation, $\theta_\ell$ is the parameter of the $\ell$-th layer and $\|\cdot\|_F$ denotes the Frobenious norm of a matrix. To solve the inner maximization problem in (1), we utilize the projected gradient descent (PGD) [32], similar to Wu et al. [55]. We remark that our pre-training uses Gaussian noise for input perturbation, while Wu et al. [55] uses a bi-level worst-case input perturbation. We found that using the Gaussian noise leads to a much more stable training for our problem. We provide more details of this optimization procedure in the Appendix A.

Such an input-level Gaussian smoothing, *i.e.*, augmenting inputs with additive noise sampled from a Gaussian distribution, has shown to effectively regularize the smoothness of DNNs in various domains [27, 30] and moreover improve the adversarial robustness [7]. Furthermore, training the model under the worst-case perturbation of weights induces a flat loss landscape with respect to weights, which has shown to be beneficial for generalization of the training objective outside the training dataset [6, 10, 21, 26, 36]. In particular, Stutz et al. [49] and Wu et al. [55] empirically observed a strong correlation between generalization capability of the model robustness at adversarial examples and the flat loss landscape of the model parameters.

## 2.3 Iterative proxy model adaptation

Once the proxy model is pre-trained to approximate the true objective function, one may apply gradient-based updates to the intermediate solution $x^{(t)}$ for time-steps $t = 0, \ldots, T - 1$, *e.g.*, $x^{(t+1)} = x^{(t)} + \eta \nabla_x f(x; \boldsymbol{\theta})|_{x=x^{(t)}}$. However, during the pre-training stage, we apply Gaussian smoothing to the proxy model only for the training dataset $\mathcal{D}$; the gradient-based updates may be erroneous for intermediate solutions far away from the training dataset $\mathcal{D}$, *i.e.*, inputs where the model is not smooth. To circumvent this issue, at each time step $t$, we adjust the output of the proxy model to be smooth at the intermediate solution $x^{(t)}$ before update these candidates. Such a model-adjustment procedure effectively focuses the model's limited expressive power for the input $x^{(t)}$. This strategy can also be interpreted as a test-time adaptation procedure [54] which adapts to information of the input $x^{(t)}$ before estimating the true objective function $f^*(x^{(t)})$.

Formally, at time-step $t$, we search for the model parameter $\boldsymbol{\theta}_t$ at the current solution candidate $x^{(t)}$ with the following objective, where the objective is optimized via the PGD method similar to the inner-maximization problem of (1):

$$\boldsymbol{\theta}_t = \arg\min_{\widetilde{\boldsymbol{\theta}} \in \mathcal{B}(\boldsymbol{\theta})} \left[ \|\nabla_x f(x; \widetilde{\boldsymbol{\theta}})\|_2 \Big|_{x=x^{(t)}} + \alpha \left( f(x^{(t)}; \widetilde{\boldsymbol{\theta}}) - f(x^{(t)}; \boldsymbol{\theta}_{t-1}) \right)^2 \right]$$

where we let $\boldsymbol{\theta}_{-1} = \boldsymbol{\theta}$ and $\alpha > 0$ is the hyperparameter to control the output of the proxy model changing too much from the regularization. We also constrain the model parameter $\boldsymbol{\theta}_t$ to be in the

local neighborhood $\mathcal{B}(\boldsymbol{\theta})$ of the original pre-trained parameter. This adjustment allows the proxy model to maintain a good approximation for the true objective function since the corresponding regression loss is lower-bounded for $\widetilde{\boldsymbol{\theta}} \in \mathcal{B}(\theta)$ from the training objective in (1). Note that we smooth the proxy model based on minimizing the input gradient norm [1, 9, 14, 47] rather than Gaussian smoothing as in Section 2.2; we found such a regularization to be more effective as it directly regularizes the gradient used for updating the solution. Here, one may ask why only the current intermediate solution $x^{(t)}$ is considered for the adjustment, rather than incorporating other inputs, *e.g.*, the previous solution $x^{(t-1)}$. This is because we are interested in *local behavior* at the $x^{(t)}$ for updates, so only considering $x^{(t)}$ to force the local smoothness is enough for the adjustment.

## 2.4 Other algorithmic components

**Discrete inputs.** Since our framework uses gradient-based updates on continuous-valued inputs, it is non-trivial to extend for discrete-valued inputs. To resolve this issue, we propose to use a variational autoencoder (VAE) [24] which maps a discrete-valued input $x$ to a continuous-valued latent vector $z$. To be specific, we train an encoder network $g(x)$ and a decoder network $h(z)$ using the VAE objective. Next, we train a proxy model $f(z; \boldsymbol{\theta})$ which operates on the latent space to approximate the true objective function, *i.e.*, $f \circ g(x) \approx f^*(x)$. Then we can optimize over the space of latent vectors using RoMA with respect to the proxy model $f(z; \boldsymbol{\theta})$. After optimizing the latent vector for $T$ steps to obtain $z^{(T)}$, we sample from the decoder $h(z^{(t)})$ to report the discrete-valued solution.

**Hyperparameter selection strategy.** Offline MBO algorithms aim to generate highly scoring solutions under the constraint of not accessing the true objective function. Unfortunately, there is no rule of thumb for selecting hyperparameters in such an offline setting [53], and it is an active area of research. Nevertheless, we provide guidelines for selecting the hyperparameters of our RoMA in the offline setting. For the maximum magnitude $\varepsilon$ of weight perturbations, we choose the largest value such that the pre-training of the proxy model in Section 2.2 is stable. For choosing $T$ and $\alpha$, we set it to a constant value which is task-agnostic. Indeed, we empirically observe the performance of RoMA to be robust under a different choice of such constant values.

# 3 Related work

**Offline model-based optimization.** Offline model-based optimization (MBO) has recently gained interest for objective functions that are expensive or dangerous to evaluate. To be specific, existing offline MBO methods generate candidate solutions based on either sampling from a generative model or gradient-based updates to a sample existing in the training dataset. For the former approach, Brookes et al. [5], Fannjiang and Listgarten [8] train variational auto-encoders [24] along with the proxy model to designate a trust-region where the optimized solution via generation under such region is accurate. Moreover, Kumar and Levine [25] learns an inverse mapping from the scores to the inputs via conditional generative adversarial network [34]. However, these approaches require a specific tuning across different domains to learn a valid manifold of high-scoring inputs.

Instead, several works have focused on the latter approach of utilizing gradient-based updates to optimize the solution. Specifically, Fu and Levine [11] maximizes the normalized maximum likelihood estimate of the true objective function, and Trabucco et al. [52] regularizes the proxy model to assign lower scores to the sample-generating policy. However, in these gradient-based methods, the non-smooth nature of deep neural networks has been overlooked. We directly tackle this issue by regularizing the smoothness of the model via proposing a two-stage framework.

**Regularizing smoothness of deep neural networks.** Smoothness regularization to a deep neural network (DNN) has been regarded as a trustworthy regularization to enhance its generalization capability in a wide range of domains [38, 57], *e.g.*, image classification [31, 47], adversarial robustness [7, 32, 56, 58], generative models [1, 9, 14, 35], and uncertainty estimation [29]. They have employed various forms of regularization that to mitigate the highly non-smooth nature of DNNs. For instance, Loshchilov and Hutter [31], Gulrajani et al. [14] suggest to directly regularize the weights of the DNN and Sokolić et al. [47] proposes to minimize the spectral norm of the DNN parameter. Moreover, Arbel et al. [1], Fedus et al. [9], Sokolić et al. [47] give an input gradient penalty while training. Another line of works, especially under the context of the adversarial robustness in image classifier, they usually attempt to regularize the smoothness by considering an input-level noise,

like Gaussian [7] or worst-case noise [32, 58] that maximizes the training objective. Inspired by these success, our method employs the smoothness prior into the offline MBO for better generalization of the model with its robustness to adversarial inputs.

**Test-time adaptation.** Test-time adaptation has shown considerable improvement to handle the distributional shift from the training data distribution at test-time [2, 18, 50, 54]. These approaches propose an objective to adapt the model at test-time to perform better at the specific task under unseen data, *e.g.*, robust image classification against corruptions [19] or generalization of reinforcement learning policy across different environments [18]. Similarly, we design a model adaptation objective so that the update of the given new solution candidate becomes robust.

## 4 Experiments

We verify the effectiveness of our framework on Design-bench [53],[2] an offline model-based optimization (MBO) benchmark consisting of 6 tasks with various domains. Our result exhibit that the proposed robust model adaptation (RoMA) framework significantly improves the overall performance in diverse offline MBO tasks compared to recent offline MBO frameworks [5, 8, 11, 25, 52]. We also perform ablation studies to validate each component in our RoMA.

We follow the same setup for all experiments in prior works for the evaluation [11, 52, 53]: we optimize and propose 128 solution candidates as the final solution and evaluate their 100th/50th percentile scores. We report the numbers obtained by the previous works unless otherwise specified. All the numbers are averaged over 16 runs along with the standard deviations for reporting our results.

**Baselines.** We consider 6 prior approaches in offline MBO as baselines for comparison: CbAS [5], Autofocus [8], NEMO [11], MINs [25], COMs [52], and naïve gradient ascent (Grad. Ascent) [53].

**Implementation details.** We use a 3-layer multi-layer perceptron (MLP) in all experiments, with the width size 64 and the softplus activation function. Adam optimizer [23] with the learning rate of 0.001 is utilized to pre-train the proxy model with the dataset of each task. Gradients are clipped by a norm of 1.0, and we set the mini-batch size as 128. For variational auto-encoder (VAE) [24], we use the same architecture as the one used in prior works to utilize VAE for offline MBO [5, 8]. For the optimization of solutions, we select inputs that have the highest score from the dataset as initial candidates, like prior works that utilize gradient-based methods to optimize solutions [11, 52]. For optimizer to update solution candidates, we also utilize Adam optimizer [23]. All the experiments are processed with 4 GPUs (NVIDIA RTX 2080 Ti) and 24 instances from a virtual CPU (Intel Xeon Silver 4214 CPU @ 2.20GHz), and it takes at most ~4 hours to run each task over 16 runs.

**Hyperparameters.** For the main experiments, we set the number of solution update to be large enough, *i.e.*, $T = 300$. For the coefficient for regularization $\alpha$, we set $\alpha = 1$ across all tasks. We choose the largest maximum magnitude of a weight perturbation $\varepsilon$ so that the pre-training of the proxy model is possible. Specifically, we set $\varepsilon = 0.0005$ for GFP, Molecule, and Superconductor task and choose $\varepsilon = 0.005$ at other 3 tasks. For the step size $\eta$, we decay $\eta$ proportional to the output of the model at the current solution from the initial value; See the Appendix B for more details.

**Tasks.** In what follows, we describe the tasks from offline MBO benchmark Design-Bench [53].

- The objective of **green fluorescence protein (GFP)** is to find a protein that has a high fluorescence, which is proposed by Sarkisyan et al. [41]. The dataset consists of a total 5000 number of proteins with fluorescence values, and each protein is composed of 238 amino acids, *i.e.*, 238 discrete variables where each variable indicates 20 categorical one-hot vectors.

- In the **Molecule** task, one finds a substructure of a molecule that has high activity at the test against the target assay [13]. The dataset involves 4,216 data points, where each data consists of 1,024 discrete variables, and each variable indicates the Morgan radius 2 substructure fingerprints.

- The **Superconductor (Supercond.)** task aims to find a superconducting material with a high critical temperature, where the dataset is provided by Hamidieh [17] and consists of 21,263 data in total. Each data is an 81-dimensional vector with the corresponding critical temperature.

- In the **HopperController (Hopper)** task, one optimizes the parameter of the policy neural network to maximize the expected return on the Hopper-v2 task in OpenAI Gym [4]. Each

---

[2]https://github.com/brandontrabucco/design-bench

Table 1: Comparison of 100th percentile scores for each task. We mark the scores within one standard deviation from the highest average score to be bold.

| | Discrete domain | | Continuous domain | | | | |
|---|---|---|---|---|---|---|---|
| Method | GFP | Molecule | Supercond. | Hopper | Ant | Dkitty | Avg.[†] |
| Dataset Max | 3.152 | 6.558 | 73.90 | 1361.6 | 108.5 | 215.9 | 1.000 |
| CbAS [5] | **3.408**±**0.029** | 6.301±0.131 | 72.17±8.652 | 547.1±423.9 | 393.0±3.750 | **396.1**±**60.65** | 1.324 |
| Autofocus [8] | 3.365±0.023 | 6.345±0.141 | 77.07±11.11 | 443.8±142.9 | 386.9±10.58 | 376.3±47.47 | 1.286 |
| NEMO [11] | 3.359±0.036 | 6.682±0.209 | **127.0**±**7.292** | 2130.1±506.9 | 393.7±6.135 | **431.6**±**47.79** | 1.687 |
| MINs [25] | 3.315±0.029 | 6.508±0.236 | 80.23±10.67 | 746.1±636.8 | 388.5±9.085 | 352.9±38.65 | 1.304 |
| COMs [52] | 3.305±0.024 | **6.876**±**0.128** | 110.0±6.804 | **2395.7**±**561.7** | 378.8±10.01 | 341.4±28.47 | 1.589 |
| Grad. Ascent [53] | 2.894±0.001 | 6.636±0.066 | 89.64±9.201 | 1050.8±284.5 | 399.9±4.941 | **390.7**±**49.24** | 1.237 |
| RoMA (Ours) | 3.357±0.024 | **6.890**±**0.122** | 103.9±5.487 | **2466.5**±**359.2** | **468.5**±**12.68** | 384.3±51.68 | **1.705** |

Table 2: Comparison of 50th percentile scores for each task. We mark the scores within one standard deviation from the highest average score to be bold.

| | Discrete domain | | Continuous domain | | | | |
|---|---|---|---|---|---|---|---|
| Method | GFP | Molecule | Supercond. | Hopper | Ant | Dkitty | Avg.[†] |
| Dataset Max | 3.152 | 6.558 | 73.90 | 1361.6 | 108.5 | 215.9 | 1.000 |
| CbAS [5] | **3.269**±**0.018** | 5.472±0.123 | 32.21±7.255 | 132.5±23.88 | 267.3±16.55 | 203.2±3.580 | 0.826 |
| Autofocus [8] | 3.216±0.029 | 5.759±0.158 | 31.57±7.457 | 116.4±18.66 | 176.7±59.94 | 199.3±8.909 | 0.752 |
| NEMO [11] | 3.219±0.039 | 5.814±0.092 | **66.41**±**4.618** | 390.2±43.37 | 326.9±5.229 | 180.8±34.94 | 0.960 |
| MINs [25] | 3.135±0.019 | 5.806±0.078 | 37.32±10.50 | 520.4±301.5 | 184.8±29.52 | 211.6±13.67 | 0.803 |
| Grad. Ascent [53] | 2.894±0.000 | **6.401**±**0.186** | 54.06±5.06 | 185.0±72.88 | 318.0±12.05 | **255.3**±**6.379** | 0.862 |
| RoMA (Ours) | 3.230±0.015 | 6.160±0.018 | **68.99**±**6.687** | **560.1**±**22.47** | **370.8**±**6.771** | 252.4±5.167 | **1.103** |

    datapoint is a pair of policy neural network weights of 5,126 parameters and its expected return simulated by a simulator from Brockman et al. [4], and the size of the dataset is 3,200.

- The goal of **AntMorphology (Ant)** and **DkittyMorphology (Dkitty)** tasks is to find the optimal robot parameters, *e.g.*, positions and orientation of robot joints. Each input is a 56 and 60-dimensional vector and the dataset size is 12,300 and 9,546, respectively.

Following prior offline MBO methods [11, 52] that have relied on Design-bench for evaluating their algorithms, we remark that our experiments are done on Design-bench, which consists of the set of tasks using MLP. Nevertheless, we emphasize that Design-bench still includes diverse scenarios (discrete and high-dimensional inputs) with important applications (biology, chemistry, material design, and robotics), which is enough to solve general MBO problems.

### 4.1 Main results

In Table 1 and 2, we report the 100th and 50th quantile scores of the solutions found by our RoMA and the baselines, respectively. We also report the average of normalized score across 6 tasks, *i.e.*, average of $\frac{y - y_{\min}}{y_{\max} - y_{\min}}$, where $y$, $y_{\max}$, $y_{\min}$ indicates the score of proposed solution from our RoMA framework, maximum and minimum values in the dataset, respectively.

As reported in Table 1, RoMA outperforms the AntMorphorlogy task with a large margin and shows a state-of-the-art result at Molecule, HopperController, and DkittyMorphology tasks. Moreover, we note that RoMA achieves at least runner-up results at all 6 tasks, *i.e.*, our framework shows the competitive results at all tasks to other prior approaches. Finally, our method shows the state-of-the-art result compared to prior works on an average of normalized scores.

Intriguingly, RoMA also shows consistent improvements and shows competitive results on both of discrete tasks, namely Molecule and GFP. Here, we note that most prior methods suffer to achieves high 100th percentile scores at both tasks. It confirms how the VAE-based mapping from discrete inputs to latents is effective for leveraging gradient-based offline MBO methods to discrete inputs.

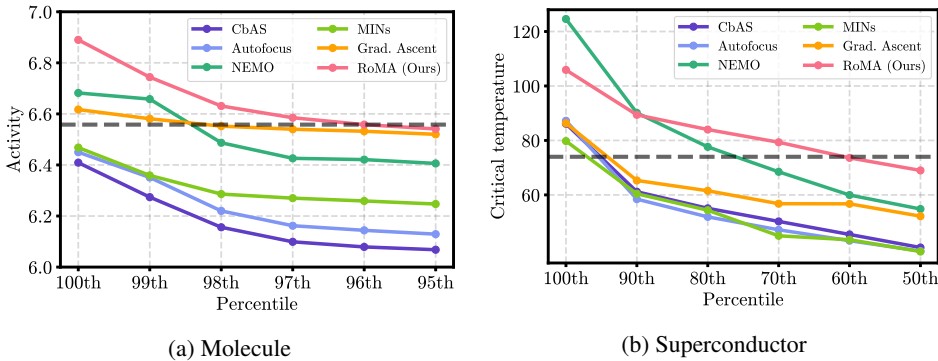

(a) Molecule

(b) Superconductor

Figure 3: Scores of the ground-truth objective function evaluated at samples of different percentiles in Molecule and Superconductor task. The dotted lines indicate the maximum score in the dataset.

We also note that RoMA dramatically improves 50th percentile scores compared to existing approaches, as shown in Table 2; our framework outperforms all at 50th percentile scores at 4 out of 6 tasks and is at the second place for the remaining tasks. Moreover, the 50th percentile scores of 3 out of 6 tasks are larger than the maximum value in the dataset, which supports the superiority of our proposed method at reliably improving the solutions using gradient-based updates.

## 4.2 Ablation studies

Finally, we perform ablation studies on our framework to validate the effect of each component. To be specific, we start by analyzing the ratio of found solutions that outperforms the best output in the dataset. We then verify the effect of each component of RoMA, namely (a) the pre-training strategy (b) and the model adjustment procedure. In the end, we conduct experiments to show the stability of the framework to a different choice of hyperparameters.

**Ratio of high-scoring solutions.** One can apply the naïve baseline of selecting the highest-scoring input in the training dataset for our optimization problem as the solution. Accordingly, it is worthwhile to analyze the ratio of solutions from each algorithm that outperform such a naïve baseline. To this end, in Figure 3, we provide scores corresponding to various percentiles for the Superconductor task and the Molecule task. Since prior works have only reported 100th/50th percentile scores for each task, we re-implemented and report other percentile scores for comparison.

In this respect, we extensively figure out the ratio of those outperforming solutions among entire candidates moreover to the 50th percentile score. Results in Figure 3 demonstrates RoMA better optimizes solution candidates to other previous approaches with the higher ratio. Precisely, the 70th percentile score in Superconductor task and the 96th percentile score in Molecule task from our framework is still larger than the maximum score from the dataset, while existing approaches fails to optimize the solution at such a percentile.

**Effect of each component.** We start with ablating the effectiveness of each component in RoMA. To this end, we compare RoMA with (a) RoMA without the model adaptation scheme and (b) RoMA without both the model adaptation and the robust pre-training schemes. To be specific, (a) pre-trains

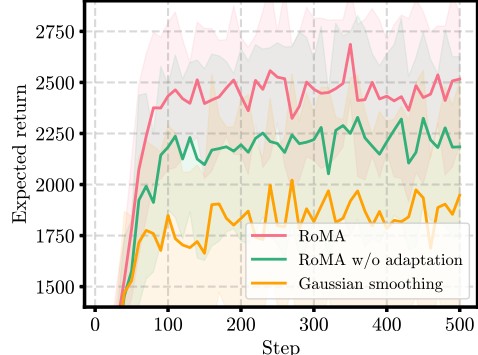

Figure 4: Illustration of an ablation study on HopperController task to analyze the behavior of each component in our framework.

the proxy model using the pre-training strategy but optimizes the solutions without the model adaptation. Next, (b) trains the proxy model only using the input-level Gaussian smoothing.

Figure 4 summarizes the result of the comparison for the Hopper task. Remarkably, replacing the naïve pre-training into our proposed strategy for pre-training significantly boosts the performance. Employing the model adjustment gives an additional improvement. This confirms how the pre-

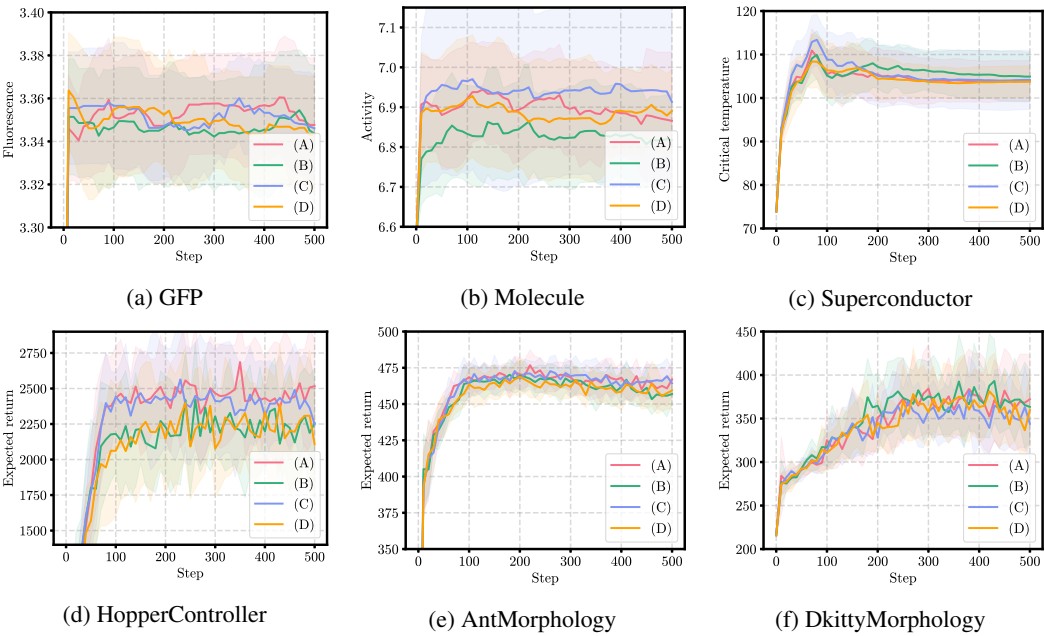

(a) GFP      (b) Molecule      (c) Superconductor

(d) HopperController      (e) AntMorphology      (f) DkittyMorphology

Figure 5: Comparison of 100th percentile scores to different hyperparameter choices in all 6 tasks. All experiments are averaged over 16 runs. Here, (A), (B), (C), and (D) indicates the result from different choices of hyperparameters $(\alpha, \varepsilon)$: $(\alpha_0, \varepsilon_0), (\alpha_0, 0.2\varepsilon_0), (0.1\alpha_0, \varepsilon_0)$, and $(0.1\alpha_0, 0.2\varepsilon_0)$, respectively. Here, $\alpha_0$ and $\varepsilon_0$ denote the hyperparameters used in our main experiments, *e.g.*, Table 1.

training and the adjustment strategies bring orthogonal improvements in our framework. We also provide an ablation study on each term of the adaptation objective; See Appendix E for details.

**Hyperparameter selection.** We remark the challenge in offline MBO to select the hyperparameter, as we do not have any access to the objective function. Accordingly, analyzing the effect of hyperparameter choices in the offline situation is crucial and worthwhile. In this respect, we empirically verify the effectiveness of our framework regarding hyperparameter choices. Specifically, we show our framework is stable at a different choice of the number of iteration $T$ and the coefficient $\alpha$ at the adjustment. Moreover, we also validate the strategy for selecting the maximum weight perturbation magnitude $\varepsilon$. To be specific, we provide curves for each task with $T = 500$, to validate the stability of our framework to the more number of iterations. For regularization coefficient $\alpha$ at the model adjustment, we compare the result of $\alpha = 0.1$ and $1$ in all tasks. In the end, we compare the result from the $\varepsilon$ determined from our procedure and the smaller magnitude $0.2\varepsilon$.

Figure 5 illustrates the result. We first note that the performance on all tasks is stable even if we choose larger $T$ for optimizing the solution. In the case of $\alpha$, the result is quite stable at the different choice of $\alpha$, which verifies choosing universal $\alpha = 1$ is enough in our framework. Moreover, the result is better or at least similar at choosing the largest $\varepsilon$ compared to the one with smaller $\varepsilon$, confirming our hyperparameter selection strategy on finding $\varepsilon$ from the offline static dataset. Here, we emphasize that we do not select hyperparameters with the best performance; we use a systematic strategy for hyperparameter selection that only relies on observations made during training. Consequently, the performance can be even better with different hyperparameters on several tasks; See Appendix E.

## 5 Discussion and conclusion

We propose a new framework for offline model-based optimization (MBO) using deep neural networks (DNNs) called robust model adaptation (RoMA). Specifically, we derive a two-step procedure based on employing a local smoothness prior to alleviate the fragility of the DNNs outside the training data, as the optimization procedures cause a domain shift from the given dataset. We emphasize that this approach is the straightforward regularization to handle such a brittleness of the model. Extensive experimental results demonstrate the superiority of algorithms across a variety of tasks, both at the best solution and the overall quality of whole solution candidates.

**Limitations and future works.** Although RoMA shows its effectiveness on design problems in the offline MBO benchmark, which contains a wide range of domains [53], we note that the evaluation of optimized solutions is not accomplished by actual experiments. Instead, the assumed ground-truth function utilized for the evaluation is synthetic, *e.g.*, robot simulator or pre-trained DNNs with a large number of the dataset. As the world of nature is often more complicated, exploiting our framework in real-world situations may face unexpected challenges and not be suitable, unlike synthetic tasks. However, we believe RoMA can still be beneficial even in these cases, similar to the success in reinforcement learning to build a framework in the simulator-based environment and successfully apply such an algorithm to actual robots [16]. Still, verifying and employing the framework in more realistic, practical situations with domain experts is a worthwhile and intriguing future research direction, and we leave it for future work. Finally, we remark that the evaluation of RoMA is done using multi-layer perceptron; applying our approach into more complicated architecture and data is also an interesting future direction, *e.g.*, image optimization with convolutional network architectures.

**Negative social impacts.** The automatic design of objects can serve a crucial role in mitigating diverse safety and economic issues in real situations, *e.g.*, aircraft design or drug discovery. Although most prior works have focused on its positive aspects, we note that such a role may be either malicious: if the desired property is the one that hurts individuals. For instance, the proposed framework can be utilized for automatically optimizing biochemicals that threaten people; it shares a similar input space to search molecules or proteins for drug discovery, but only the purpose is on the opposite side. In this respect, one should be aware of the drawback of improving offline MBO algorithms.

## Acknowledgments and Disclosure of Funding

SY thanks anonymous reviewers, Jaeho Lee, Sangwoo Mo, Soojung Yang, Seokhyun Moon, and Jaeyoung Park for their helpful feedback on the early version of the manuscript. This work was partially supported by Institute of Information & Communications Technology Planning & Evaluation (IITP) grant funded by the Korea government (MSIT) (No.2019-0-00075, Artificial Intelligence Graduate School Program (KAIST)). This work was mainly supported by Samsung Research Funding & Incubation Center of Samsung Electronics under Project Number SRFCIT1902-06.

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
