# A Detailed description of the projection gradient ascent

In this section, we further describe details on how the training objective in Section 2.2 is optimized. For the inner maximization problem, we utilize projected gradient descent (PGD) [32] method to find perturbed weights $\widetilde{\boldsymbol{\theta}}$ like in [55]. Specifically, we iterate the following step for $M > 0$ times at each update of model parameters:

$$\phi_\ell \leftarrow (\widetilde{\theta}_\ell - \theta_\ell) + \gamma_1 \frac{\nabla_{\widetilde{\theta}_\ell} \mathcal{L}(\boldsymbol{\theta})}{\|\nabla_{\widetilde{\theta}_\ell} \mathcal{L}(\boldsymbol{\theta})\|_F} \cdot \|\theta_\ell\|_F, \ \mathcal{L}(\boldsymbol{\theta}) = \left[ \mathbb{E}_{(x,y)\sim\mathcal{D}, \delta\sim\mathcal{N}(0,\sigma)} \left[ (f(x+\delta; \widetilde{\boldsymbol{\theta}}) - y)^2 \right] \right]$$

$$\widetilde{\theta}_\ell \leftarrow \theta_\ell + \min(\|\phi_\ell\|_F, \varepsilon \cdot \|\theta_\ell\|_F) \cdot \frac{\phi_\ell}{\|\phi_\ell\|_F},$$

for $\ell = 1, \ldots, L$ where $\gamma_1 > 0$ is a hyperparameter denoting the step size for the inner maximization problem and define $\widetilde{\theta}_\ell := \theta_\ell$ for $l = 1, \ldots, L$ at the first time. We let $\gamma_1 = \frac{\varepsilon}{M}$ and $M = 20$ for all experiments. The objective in Section 2.3 is optimized similarly with $M = 100$.

After finding perturbed model weights $\theta_\ell$, we optimize the model parameter $\theta_\ell$ by the following update for $\ell = 1, \ldots, L$:

$$\theta_\ell \leftarrow \widetilde{\theta}_\ell - \gamma_2 \nabla_{\widetilde{\theta}_\ell} \left[ \mathbb{E}_{(x,y)\sim\mathcal{D}, \delta\sim\mathcal{N}(0,\sigma)} \left[ (f(x+\delta; \widetilde{\boldsymbol{\theta}}) - y)^2 \right] \right] - (\widetilde{\theta}_\ell - \theta_\ell),$$

which is rewritten as follows:

$$\theta_\ell \leftarrow \theta_\ell - \gamma_2 \nabla_{\theta_\ell} \left[ \mathbb{E}_{(x,y)\sim\mathcal{D}, \delta\sim\mathcal{N}(0,\sigma)} \left[ (f(x+\delta; \widetilde{\boldsymbol{\theta}}) - y)^2 \right] \right].$$

Here, $\gamma_2$ is a learning rate for optimizing $\boldsymbol{\theta}$, which is set to $0.001$ in all experiments.

# B  More description on implementation details

**Data split.** We utilize a random 500 number of data as the validation set of the Superconductor task and 200 data for the rest of the 5 tasks.

**Variational auto-encoder architecture.** We use the same variational auto-encoder [24] architecture in another offline model-based optimization (MBO) algorithms [5, 8], which like in Trabucco et al. [53]. Specifically, the encoder and the decoder network are a 2-layer multi-layer perceptron (MLP), where the activation function is a leaky-relu function. Here, a dimension of a hidden dimension and a latent dimension is set to 50 and 32, respectively. We generate and propose the solution via the `argmax` operator from the optimized latent vector and the decoder network as the final solution candidate for a fair comparison to other gradient-based optimization baselines.

**Step size $\eta$.** For step size $\eta$ for solution updates, we set $\eta = 0.002$ for GFP, Molecule, and HopperController task, and use $\eta = 0.003$ for other 4 tasks. We decay the step size $\eta$ for each update based on the concept of "trust-region" to prevent the generation of solution at infeasible regions. To be specific, we set the $t$-th step size $\eta_t$ at the current candidate $x^{(t)}$ as $\eta_t := \eta_0(1 - \frac{1}{N\sigma}(f(x^{(t)}; \boldsymbol{\theta}_t) - y^{(0)}))$. This encourages the solution to stay inside the trust-region $\{x \in \mathbb{R}^L : f(x; \boldsymbol{\theta}) \leq y^{(0)} + N\sigma\}$ where $\sigma$ is the standard deviation of outputs in the dataset $\mathcal{D}$, and $N$ is a reasonably large enough constant, *e.g.*, $N \geq 4$. We found such a design choice yields a robust behavior of our framework across all tasks.

# C  Description for offline model-based optimization baselines

In this section, we breifly describe the algorithms we used as offline model-based optimization baselines for evaluating our framework at a high level.

- **CbAS** [5] trains a variational auto-encoder (VAE) [24] to optimize the solution, along with the pre-trained proxy model. Specifically, such a VAE model gradually focuses on the optimal solution from updating the VAE model, starting from the whole input space from the given dataset.

- **Autofocus** [8] is a modified version of CbAS, which also re-trains the proxy model while adapting the VAE. By this re-training, the proxy model less suffers from the distributional shift on this VAE adaptation procedure.

- **NEMO** [11] approximates the normalized log-likelihood (NML) via fine-tuning of proxy models using quantized output labels to alleviate the adversarial examples from gradient-based updates. They update the solution to maximize this NML estimation.

- **MINs** [25] learn an inverse mapping from the score to the corresponding output. Specifically, they train a conditional generative adversarial network (cGAN) [34] to achieve this. After that, they optimize the output score used for querying the trained inverse map so that the optimal solutions are generated by the sampling from this inverse mapping.

- **COMs** [52] train a proxy model with a conservative objective to mitigate the overestimation from a gradient-based optimization. To be specific, they additionally consider the regularization so that the output of gradient-based updated inputs is conservative from the original input.

- **Grad. Ascent** [53] trains an ensemble of proxy models and performs naïve gradient ascent starting from high-scoring inputs to optimize the solution.

# D  Detail of tasks used for evaluation

We notice that it is impossible to access the ground-truth function for evaluation as it often accompanies real experiments. In this respect, alternative reasonable oracle function foe evaluation is required. In this section, we describe those details on how Trabucco et al. [53] designs oracle function to evaluate the optimized solution, which is also used to compute the score of our results, *e.g.*, Table 1. Moreover, we also introduce how the data is collected.

**GFP.** The evaluation is done by computing the mean score of the oracle function proposed by Brookes et al. [5], which is an Gaussian process regression model. Here, the protein-specific kernel is utilized, which is from Shen et al. [44]. The dataset is collected from physical experiments to measure the fluorescence of the protein.

**Molecule.** The oracle function used in Molecule task is a random forest regression model trained with the whole data, which is proposed by Martin et al. [33]. Here, the dataset is collected from real experiments and top 20th percentile data are removed to ensure the offline model-based optimization framework to optimize new outputs that has the higher score than given dataset, but can be accurately evaluated from the random forest model.

**Superconductor.** The oracle function in this task is a random forest model using the whole set of data. For the data used as the static dataset for our offline model-based optimization task, the top 20th percentile of the original dataset is removed, and only the rest of the 80th percentiles are used so that high-scoring inputs are not visible during the search of solutions. The total dataset that is used for training the random forest model is public, collected by Hamidieh [17].[3]

**HopperController.** As mentioned in Section 4, inputs in the HopperController task are neural weights of the policy neural network. Accordingly, the oracle function evaluates the policy network under the Hopper-v2 environment in the MuJoCo simulator.[4] Specifically, the sum of rewards over 1,000 time-steps becomes the output score of the HopperController task at given input. For collecting the dataset, Trabucco et al. [53] trains 32 independent policy neural networks under Proximal Policy Optimization (PPO) algorithm [42] over 10 million steps, where weights are saved at every 10,000 steps during the training. The output label is identically determined to the rule of evaluating inputs.

**AntMorphology and DkittyMorphology.** Inputs of these tasks are the morphology of each robot. The exact evaluation is achieved via a pre-trained policy neural network using soft actor-critic (SAC) [15] trained for 3 million steps in MuJoCo Ant and ROBEL D'kitty environment under the fixed morphology. Specifically, the score is determined as the average over 16 rollouts from the MuJoCo simulator and this pre-trained policy neural network. For the dataset collection, morphology with the original value and Gaussian noise is collected, with the corresponding evaluation outputs.

---

[3] https://archive.ics.uci.edu/ml/datasets/Superconductivty+Data
[4] http://www.mujoco.org

# E   More results on ablation studies

**Effect of each adaptation term.**  Note that our proposed adaptation objective (in Section 2.3) is composed of two terms: regularizing a gradient norm (first term) and consistency of outputs (second term). To verify their effect, we provide experimental results when only one of the two terms is used. As shown in Table 3, both terms are necessary to regularize the proxy model appropriately.

Table 3: Comparison of 100th percentile scores if only one of two terms are used for the adaptation.

|  | Hopper |
| --- | --- |
| Ours (both terms) | **2466.5±359.2** |
| Gradient norm (first term) | 2183.4±329.6 |
| Consistency (second term) | 2081.9±414.4 |

**Quantitative results of different hyperparameters.**  We also provide 100th percentile scores achieved by different hyperparameter setups of the coefficient $\alpha$ and the perturbation magnitude $\varepsilon$, where the number of iteration $T$ is set to be equal to main experiments, *i.e.*, $T = 300$.

Table 4: Comparison of 100th percentile scores for each task with different hyperparmaeters. We mark the highest scores to be bold. Here, $\alpha_0$ and $\varepsilon_0$ denote the hyperparameters used in our main experiments, *e.g.*, Table 1.

| Hyperparam. | Discrete domain | | Continuous domain | | | |
| --- | --- | --- | --- | --- | --- | --- |
|  | GFP | Molecule | Supercond. | Hopper | Ant | Dkitty |
| $(\alpha_0, \varepsilon_0)$ | **3.357±0.024** | 6.890±0.122 | 103.9±5.487 | **2466.5±359.2** | 468.5±12.68 | **384.3±51.68** |
| $(0.5\alpha_0, \varepsilon_0)$ | 3.351±0.022 | 6.893±0.151 | 104.7±5.384 | 2420.5±447.0 | **469.8±10.68** | 378.6±46.92 |
| $(0.1\alpha_0, \varepsilon_0)$ | 3.351±0.023 | **6.941±0.219** | 104.1±6.655 | 2412.3±365.8 | 469.1±10.39 | 358.2±27.67 |
| $(\alpha_0, 0.2\varepsilon_0)$ | 3.342±0.026 | 6.823±0.134 | **106.3±4.706** | 2303.6±301.7 | 465.2±11.12 | 370.6±40.77 |
| $(0.1\alpha_0, 0.2\varepsilon_0)$ | 3.350±0.023 | 6.871±0.154 | 103.7±3.797 | 2088.1±348.5 | 461.8±8.389 | 357.3±41.76 |

Table 4 illustrates the result. Although the hyperparameter choice under our proposed strategy shows reasonable performance across all tasks, one can show the performance can be even better with different hyperparameter selections, *e.g.*, Molecule and Superconductor tasks.