# OpenReview forum: "RoMA: Robust Model Adaptation for Offline Model-based Optimization"
_NeurIPS.cc/2021/Conference — NeurIPS 2021 Poster_

### Official Review · Reviewer_3o7x · 2021-07-02

**Rating:** 6
**Confidence:** 4

**Summary:**

This paper proposes a gradient-based method that searches for inputs that maximizes the output of a (black-box) score function (model-based optimization).   In the analyzed scenario, the score function is an adversarially-robust pre-trained neural network, and the searching algorithm is gradient ascent (wrt to inputs) with a local regularizer in terms of the model parameters.

**Limitations And Societal Impact:**

Yes

**Main Review:**

Pros.

- The proposed method achieves good empirical results in continuous tasks in the Design-Bench benchmark.
- The authors proposed solutions for both continuous and discrete inputs.


Cons.

My major concern is about the technical novelty and experimental setting/scope of the work.

- Wrt to novelty, the pre-training strategy follows from existing work in the context of adversarial robustness. The iterative proxy model adaptation is "gradient ascent" with a regularizer that updates the model parameters based on a hand-designed objective. I appreciate (and agree up to some extent with)  the intuitive motivation provided for this objective. However, many other regularizers can be proposed based on a similar intuition. Since this is the major technical contribution of the work, and the experimental section considers only a single benchmark with a 3 layers MLP, I believe a better theoretical motivation/justification could be provided.

- Experimental Setting/Scope: Many tasks can be formulated in the scope introduced in section 2.1 (line 74-79) some very challenging  (e.g. class-conditional image generation).  However, it is not clear why the tasks in the chosen benchmark should be considered over others. Would the proposed framework (RoMA) still be effective in more complex tasks and networks? If only such a benchmark is considered then the scope of the paper should be narrowed and motivation for the more narrowed scope should be provided.
- Writing. I consider several parts of the writing that can be improved e.g. introduction and abstract could be improved.

Questions:

- How important is regularizing the gradient norm vs keeping the output of the model close between steps? (e.g. first vs the second term in line 138) (This is not captured in figure 4).
- Why model "adaptation" instead of model adjustment? In other words, is "adaptation" the right word?  If searching starts from  x_0 ~ D, and every step is only changing x  very locally why  x_1=x_0 + delta is necessarily another distribution.

**Time Spent Reviewing:**

8

---

> ### Author Response · Authors · 2021-08-09
> **Response to Reviewer 3o7x**
>
> We sincerely appreciate your efforts and insightful comments to improve the manuscript. We respond to each of your comments one-by-one in what follows.
>
> ---
>
> **[C1] Lacking novelty since the pre-training strategy follows from existing work on adversarial robustness.**
>
> As we mentioned in Section 2.2., our pre-training strategy is similar to AWP [Wu et al., 2020] for adversarial robustness (they are not exactly the same; see our response C1 to Reviewer yVCx). However, this similarity is rather a strength of our paper: We think it is quite novel to solve the offline model-based optimization (MBO) problem via the lens of adversarial robustness. We believe such a contribution to be significant since it highlights an important connection between two important research areas which has not been known before.
>
> ---
>
> **[C2] Concerns on how “many other regularizers can be proposed based on the intuition of iterative proxy model adaptation.”**
>
> We remind you that our goal/idea here is adapting the output of the proxy model to be locally smooth at the current solution, where there can be several possibilities in technical details under the direction. We emphasize that our central contribution lies in the design of the overall  2-stage pipeline injecting “smoothness prior” for the offline MBO problem, not its technical details. While we agree that other regularizers can be designed, the proposed regularizer is the most straightforward one under our intuition; the objective directly regularizes the smoothness as well as the prediction at the current solution. Developing a better form of regularizer should be an interesting future direction.
>
> ---
>
> **[C3] Design-bench experiments lacking motivation and having a narrow scope.**
>
> We just follow the same benchmark setup considered in recent prior offline MBO works due to a fair comparison to prior offline MBO methods and reproducible research, i.e., most recent prior offline MBO works [Fu & Levine., 2021; Trabucco et al., 2021] have relied on Design-bench for evaluating their algorithms. We think Design-bench still includes diverse scenarios (discrete and high-dimensional inputs) with important applications (biology, chemistry, material design, and robotics). Hence, our experiments do not narrow down our scope from solving general MBO problems. Nevertheless, to alleviate your concern, we will explicitly state how our experiments are done on the considered set of tasks using multi-layered perceptrons in our final manuscript. Furthermore, if you could suggest a new experiment of your interest, we will incorporate this as well.
>
> ---
>
> **[C4] Several parts of the writing can be improved, e.g., introduction and abstract.**
>
> Following your suggestion, we will improve our writing, including the introduction and abstract. If you could guide us on which part of writing is particularly to be improved, it would be very much appreciated!
>
> ---
>
> **[C5] How important is regularizing the gradient norm (first term) vs. keeping the output of the model close between steps (second term)?**
>
> To answer your question, we provide experimental results when only one of the two terms is used for the regularization of the model. The results confirm that each regularization term provides an orthogonal improvement with a similar magnitude.
>
> \begin{array}{l | cc}
> & \text{Hopper} & \text{Supercond.} \newline
> \hline
> \text{Ours (both terms)} & \mathbf{2401.6}\small{\pm248.7} & \mathbf{108.6}\small{\pm8.153} \newline
> \hline
> \text{Gradient norm (first term)}   & 2183.4\small{\pm329.6} & 102.8\small{\pm6.430} \newline
> \text{Consistency (second term)} & 2081.9\small{\pm414.4} & 103.7\small{\pm6.161} \newline
> \end{array}
>
> ---
>
> **[C6] Why does changing the inputs via local updates necessarily result in another distribution? Why use the terminology “adaptation” over “adjustment”?**
>
> It has been well evidenced that the behavior of the neural network model can change dramatically even when the input distribution is changed using a few gradient-based updates, e.g., adversarial attacks [Madry et al., 2017]. Therefore, such a distribution shift has often been regarded to result in a distinct distribution in many prior works, e.g., [Lee et al., 2018; Xie et al., 2020]. Since our approach uses hundreds of gradient-based updates, we expect the distribution shift to be even larger. We hope this clarifies your concern.
>
> ---
>
> [Wu et al., 2020] Adversarial Weight Perturbation Helps Robust Generalization, NeurIPS 2020
> [Trabucco et al., 2020] Design-bench: Benchmarks for Data-driven Offline Model-based Optimization, 2020
> [Fu & Levine., 2021] Offline Model-based Optimization via Normalized Maximum Likelihood Estimation, ICLR 2021
> [Trabucco et al., 2021] Conservative Objective Models for Effective Offline Model-based Optimization, ICML 2021
> [Madry et al., 2017] Towards Deep Learning Models Resistant to Adversarial Attacks, ICLR 2018
> [Lee et al., 2018] A Simple Unified Framework for Detecting Out-of-distribution Samples and Adversarial Attacks, NeurIPS 2018
> [Xie et al., 2020] Adversarial Examples Improve Image Recognition, CVPR 2020

---

> > ### Author Response · Authors · 2021-08-24
> > **A gentle reminder for Reviewer 3o7x**
> >
> > Dear Reviewer 3o7x,
> >
> > Thank you very much again for your time and efforts in reviewing our paper.
> >
> > We kindly remind that we have only a week or so in the discussion period.
> >
> > We just wonder whether there is any further concern and hope to have a chance to respond before the discussion phase ends.
> >
> > Many thanks, Authors

---

> > > ### Comment · Reviewer_3o7x · 2021-08-26
> > > **Thanks for the detailed response and additional experimental results.**
> > >
> > > I thank the authors for the detailed response and additional experimental results.  My concern with respect to technical novelty still remains as I believe a better theoretical motivation/justification could make this paper stronger.  That said,  I have updated my score to 6 because my concern with respect to the scope  of the work has been addressed in C3, and the authors has promised to explicitly state this in the final manuscript.  The additional experimental results(C5) also shows the effectiveness of the two terms for the regularization of the model. I recommend  the authors to also add this result to the final manuscript.

---

> > > > ### Author Response · Authors · 2021-08-27
> > > > **Thank you for the response**
> > > >
> > > > We are happy to hear that our rebuttal addressed your concerns well.
> > > >
> > > > We will explicitly state the scope of our work to avoid such confusion. We will also add the additional experimental results as you suggested.
> > > >
> > > > We admit that we are very much excited about our idea, as this is completely a new approach/idea for our problem (offline model-based optimization). On the other hand, we also agree that our idea was somewhat inspired from that in the context of adversarial robustness. We will explicitly mention the relation and the difference.
> > > >
> > > > Thank you again for the valuable suggestions and comments to add, which we believe strengthen our paper.
> > > >
> > > > If you have any remaining suggestions or concerns, please let us know!
> > > >
> > > > Best, Authors.

---

### Official Review · Reviewer_XJN3 · 2021-07-15

**Rating:** 7
**Confidence:** 3

**Summary:**

RoMA is a framework that leverages the local smoothness prior to robustly approximate black-box objective functions. RoMA consists of two steps: 1, train the proxy models (neural networks) to approximate black-box objective functions by perturbing original inputs with Gaussian inputs and minimizing worst-case weight perturbations; 2, iteratively adapting the proxy model to enforce the local smoothness of loss landscape around the input x(t). This paper devises two separate loss functions for optimizing neural networks in these two steps. As a result, this method serves to avoid adversarial optimized inputs.
In the experiment section, this method is compared to the state-of-the-art methods on the Design-bench dataset. Results show that this method achieves state-of-the-art performance across a variety of tasks.


**Limitations And Societal Impact:**

This paper includes comprehensive discussions on limitations, future works, and potential negative social impacts in the last section.

**Main Review:**

Originality:
This method involves two main steps, i.e., pre-training with weight perturbations and iterative proxy model adaptation. The first step leverages techniques from adversarial training; the second step utilizes the test-time adaptation technique. The combination of these two achieves good performance. In the related works section, literature in offline model-based optimization and regularizing smoothness of deep neural networks are summarized. There are two concerns. First, it would be helpful to give literature about test-time adaptation techniques related to the second step in this method; Second, it would be better to discuss the connections between previous methods and this method explicitly. For instance, what are the shortcomings of previous methods, and how does this method solve those shortcomings?

Quality:
Overall, this paper is technically sound:
1.	The optimization objective functions devised for each step are sensible. It is clear how those objective functions could serve to achieve the desired property.
2.	This paper’s experiments are comprehensive. The comparison between this method and other baselines on the Design-bench benchmark shows that this method could outperform previous methods across various tasks.
3.	The ablation studies also demonstrate the necessity of each component in their method.
One concern is that since this method does not outperform other methods in all tasks, some analysis on why other methods in some tasks beat this method could better explain the potential limitations of this method. This paper is a complete piece of work. It presents the full idea with comprehensive experimental results to support its claims. This paper explicitly talks about the strengths, limitations, and future works in the last section. As for the weakness, considering that other methods on two Design-bench tasks outperform this method, some discussion on possible reasons for this would be appreciated. Also, besides the lacking of actual experiments limitation, more analysis on the weakness of this method is expected.

Clarity:
This paper is clearly written. The ideas behind each component are easy to understand. One concern is the term “input-level regularization”. Some brief definitions of this term would be appreciated. For the implementation details, this paper provides comprehensive implementation details in the experiment section. Also, details about each task and how they are evaluated are provided in both the paper and its supplementary materials.

Significance:
This paper achieves good performance in the black-box objective function estimation task. Their method is easy to understand. After comparing to other methods, RoMA achieves state-of-the-art results at some tasks and overall quality. As a result, other researchers and practitioners are likely to build their work on this paper.

**Time Spent Reviewing:**

11

---

> ### Author Response · Authors · 2021-08-09
> **Response to Reviewer XJN3**
>
> We sincerely appreciate your efforts and insightful comments to improve the manuscript. We respond to each of your comments one-by-one in what follows.
>
> ---
>
> **[C1] Missing references on test-time adaptation.**
>
> Thank you for your constructive suggestion. In addition to [Wang et al., 2021] in the current manuscript (L134 in Section 2.3), we will add the following relevant literature on different domains: [Sun et al., 2020; Hansen et al., 2021; Banerjee et al., 2021]. Similar to the second step of our method, these approaches propose an objective to adapt the model to perform better at the specific task under unseen data. We will add this point with more discussions in Section 3.
>
> ---
>
> **[C2] More connections to prior works.**
>
> To incorporate your comment, we will describe the following connections between our method and prior model-based optimization (MBO) algorithms.
>
> Some previous offline MBO algorithms [Brookes et al., 2019; Fanjiang & Listgarten, 2020; Kumar & Levine, 2020] have often optimized the solution via generative models, but these approaches require a specific tuning across different domains to learn a valid manifold of high-scoring inputs. Instead of relying on models to generate the solution, our method uses gradient-based updates to optimize the solution, similar to [Fu & Levine, 2021; Trabucco et al., 2021]. In the case of the previous algorithms using gradient-based updates, the non-smooth nature of deep neural networks has been overlooked. In contrast, we directly tackle this issue by regularizing the smoothness of the model via proposing a 2-stage scheme to achieve a better generalization.
>
> ---
>
> **[C3] Discussion on possible reasons for two non-outperforming tasks (GFP and Superconductor) and analysis of the weakness of our method.**
>
> For the GFP task, we note that each data is a highly structured discrete input, i.e., a sequence of 20-categorical one-hot vectors. Accordingly, applying gradient-based updates without considering the sequential nature of the problem might be less beneficial. Moreover, for the Superconductor task, we found that values of a single dimension in the dataset is highly irregular, i.e., the largest difference across the same coordinate in the dataset is 12.79 even under the input normalization process following [Trabucco et al., 2020]. This difference is relatively larger than other tasks, e.g., differences are 8.510 and 9.762 in the HopperConroller and AntMorphology tasks, respectively. Consequently, leveraging our local updates for a fixed number of steps may be less effective. We will add such discussion to the final manuscript.
>
> ---
>
> **[C4] Unclear terminology of "Input-level regularization."**
>
> We fully agree with your concern. We used this terminology to indicate regularization of the model based on perturbing the inputs via Gaussian noise. To make our terminology more precise, we will replace the terminology with “regularization based on input-level perturbation” in the final manuscript. We will also add clear definitions to explain the other new terminology.
>
> ---
>
> [Wang et al., 2021] Tent: Fully Test-time Adaptation by Entropy Minimization, ICLR 2021
> [Sun et al., 2020] Test-time Training with Self-supervision for Generalization under Distributional Shift, ICML 2020
> [Hansen et al., 2021] Self-supervised Policy Adaptation during Deployment, ICLR 2021
> [Banerjee et al., 2021] Self-supervised Test-time Learning for Reading Comprehension, NAACL 2021
> [Brookes et al., 2019] Conditioning by Adaptive Sampling for Robust Design, ICML 2019
> [Fanjiang & Listgarten, 2020] Autofocused Oracles for Model-based Design, NeurIPS 2020
> [Kumar & Levine, 2020] Model Inversion Networks for Model-based Optimization, NeurIPS 2020
> [Fu & Levine, 2021] Offline Model-based Optimization via Normalized Maximum Likelihood Estimation, ICLR 2021
> [Trabucco et al., 2021] Conservative Objective Models for Effective Offline Model-based Optimization, ICML 2021
> [Trabucco et al., 2020] Design-bench: Benchmarks for Data-driven Offline Model-based Optimization, 2020

---

> > ### Comment · Reviewer_XJN3 · 2021-08-21
> > **Keep my score**
> >
> > Thanks for the responses. I have read the responses to my review as well as to others'. I think those responses address my concerns well. As a result, I currently keep my score.

---

> > > ### Author Response · Authors · 2021-08-24
> > > **Thank you for the response**
> > >
> > > We are happy to hear that our rebuttal addressed your concerns well.
> > >
> > > Thank you again for the valuable suggestions and comments to add, which we believe strengthen our paper.
> > >
> > > If you have any remaining suggestions or concerns, please let us know!
> > >
> > > Best, Authors.

---

### Official Review · Reviewer_yVCx · 2021-07-18

**Rating:** 6
**Confidence:** 3

**Summary:**

This paper proposes robust model adaptation, which first do a pretraining with adversarial weight perturbation. Then the final result is obtained by iteratively doing model adaptation on the learned model, with a local smoothness penalty objective. The proposed results shows improvements over existing methods.

**Limitations And Societal Impact:**

The authors have described the limitations and impact in the paper.

**Main Review:**

The paper is generally easy to follow, the motivation for the added smoothness penalty is clear. The pretraining step is very similar to what is proposed in [46], it would better if the authors could clearly illustrate the differences, if any. One concern is regarding hyper-parameter selection, as the authors have mentioned and also demonstrated in section 4.2. Although the method is relatively stable on several different hyper-parameter settings, the variance on some of the datasets seems large (i.e. Superconductor, HopperController). It seems that the selected alpha and eps values are the best performing ones. It may be good to include the results of other hyper-parameter settings in the main experiments so that it would be a more clear comparison, since the performance of the proposed method and other methods are close in most of the scenarios.

**Time Spent Reviewing:**

8

---

> ### Author Response · Authors · 2021-08-09
> **Response to Reviewer yVCx**
>
> We sincerely appreciate your efforts and insightful comments to improve the manuscript. We respond to each of your comments one-by-one in what follows.
>
> ---
>
> **[C1] Difference between AWP [46] and our pre-training strategy.**
>
> As you pointed out, our pre-training strategy looks similar to AWP, while some details are different: our method uses Gaussian noise for input perturbation while AWP uses a bi-level worst-case input perturbation (we found that using the Gaussian noise leads to a much more stable training for our problem). We will clearly illustrate the difference in the final manuscript.
>
> We emphasize that one of our contributions is to adapt an idea of adversarial robustness (target of AWP) to a new problem, i.e., the offline model-based optimization problem (target of our method). We think it is quite novel to solve the offline model-based optimization (MBO) problem via the lens of adversarial robustness.
>
> ---
>
> **[C2] Selecting hyperparameters with the best performance under large variance.**
>
> We do not select hyperparameters with the best performance. As mentioned in Section 2.4 (in L159-L166), we use a systematic strategy for hyperparameter selection that only relies on observations made during training. Nevertheless, we fully agree that it would be useful to include the exact performance of our model under different hyperparameters. We plan to report the following table in the final manuscript (with additional hyperparameter setups more than those considered in Figure 5).
>
>
> \begin{array}{l | cccccc}
> \text{Hyperparam. }& \text {GFP} & \text {Molecule} & \text {Supercond.} & \text {Hopper} & \text {Ant} & \text {Dkitty} \newline
> \hline
> (\alpha_0, \varepsilon_0)
> & 3.349\small{\pm0.032} & \mathbf{6.936} \small{\pm0.237} & \mathbf{108.6} \small{\pm8.153}
> & 2401.6\small{\pm248.7} & 464.4\small{\pm13.52} & \mathbf{390.0}\small{\pm38.04} \newline
> \hline
> (0.5\alpha_0, \varepsilon_0)
> & 3.351\small{\pm0.022} & 6.893\small{\pm0.151} & 104.7\small{\pm5.384}
> & \mathbf{2420.6}\small{\pm447.0} & \mathbf{469.8}\small{\pm10.68} & 378.6\small{\pm46.92} \newline
> (0.1\alpha_0, \varepsilon_0)
> & \mathbf{3.358}\small{\pm0.032} & 6.919\small{\pm0.186} & 104.2\small{\pm5.493}
> & 2247.3\small{\pm378.0} & 461.9\small{\pm10.71} & 369.5\small{\pm43.00} \newline
> (\alpha_0, 0.2\varepsilon_0)
> & 3.354\small{\pm0.025} & 6.903\small{\pm0.127} & 105.4\small{\pm6.654}
> & 2101.0\small{\pm495.5} & 459.7\small{\pm13.97} & 362.3\small{\pm42.39} \newline
> (0.1\alpha_0, 0.2\varepsilon_0)
> & 3.348\small{\pm0.026} & 6.832\small{\pm0.152} & 104.3\small{\pm7.584}
> & 1910.9\small{\pm387.0} & 458.0\small{\pm8.603} & 363.7\small{\pm35.50} \newline
> \end{array}
>
> In the above table, $(\alpha_0, \varepsilon_0)$ denotes the hyperparameter found by our selection scheme (in L159-L166) and used for Table 1. One can indeed observe that we do not report the result with the best-performing hyperparameters (best scores are marked to be bold). For example, the 100th percentile score in the GFP task in Table 1 is 3.349, while it can reach 3.358 with a different hyperparameter $(0.1\alpha_0, \varepsilon_0)$. We will emphasize this in the final manuscript. Nevertheless, as shown in Figure 5 and the above table, our method is not too sensitive to hyperparameters, and our selection scheme suggests a reasonable choice.

---

> > ### Author Response · Authors · 2021-08-24
> > **A gentle reminder for Reviewer yVCx**
> >
> > Dear Reviewer yVCx,
> >
> > Thank you very much again for your time and efforts in reviewing our paper.
> >
> > We kindly remind that we have only a week or so in the discussion period.
> >
> > We just wonder whether there is any further concern and hope to have a chance to respond before the discussion phase ends.
> >
> > Many thanks, Authors

---

> > > ### Comment · Reviewer_yVCx · 2021-08-26
> > > **Thanks for the response**
> > >
> > > Thanks for providing additional explanation and results. I think they addressed most of my concerns and I have adjusted my score

---

> > > > ### Author Response · Authors · 2021-08-27
> > > > **Thank you for the response**
> > > >
> > > > We are happy to hear that our rebuttal addressed your concerns well.
> > > >
> > > > Thank you again for the valuable suggestions and comments to add, which we believe strengthen our paper.
> > > >
> > > > If you have any remaining suggestions or concerns, please let us know!
> > > >
> > > > Best, Authors.

---

### Author Response · Authors · 2021-08-18
**A gentle reminder**

Dear Reviewers,

Thank you for your time and efforts in reviewing our paper.

We kindly remind that we are more than one week into the discussion period. We believe that we sincerely and successfully address your concerns/questions/misunderstandings/suggestions, with the results of the supporting experiments.

If you have any further concerns or questions, please do not hesitate to let us know.

Thank you very much!
Authors

---

### Decision · Program_Chairs · 2021-09-27

**Decision:**

Accept (Poster)

**Comment:**

The paper makes a somewhat incremental, yet solid and effective contribution to the area of model-based optimization by robust pre-training and a test-time robustification process. I concur with the reviewers and recommend the paper to be accepted at NeurIPS.